# Molecular and Clinical Opposite Findings in 11p15.5 Associated Imprinting Disorders: Characterization of Basic Mechanisms to Improve Clinical Management

**DOI:** 10.3390/ijms20174219

**Published:** 2019-08-28

**Authors:** Katharina Wesseler, Florian Kraft, Thomas Eggermann

**Affiliations:** Institute of Human Genetics, University Hospital, Technical University Aachen (RWTH), 52074 Aachen, Germany

**Keywords:** genomic imprinting, Silver–Russell syndrome, Beckwith–Wiedemann syndrome, therapy

## Abstract

Silver–Russell and Beckwith–Wiedemann syndromes (SRS, BWS) are rare congenital human disorders characterized by opposite growth disturbances. With the increasing knowledge on the molecular basis of SRS and BWS, it has become obvious that the disorders mirror opposite alterations at the same genomic loci in 11p15.5. In fact, these changes directly or indirectly affect the expression of *IGF2* and *CDKN1C* and their associated pathways, and thereby, cause growth disturbances as key features of both diseases. The increase of knowledge has become possible with the development and implementation of new and comprehensive assays. Whereas, in the beginning molecular testing was restricted to single chromosomal loci, many tests now address numerous loci in the same run, and the diagnostic implementation of (epi)genome wide assays is only a question of time. These high-throughput approaches will be complemented by the analysis of other *omic* datasets (e.g., transcriptome, metabolome, proteome), and it can be expected that the integration of these data will massively improve the understanding of the pathobiology of imprinting disorders and their diagnostics. Especially long-read sequencing methods, e.g., nanopore sequencing, allowing direct detection of native DNA modification, will strongly contribute to a better understanding of genomic imprinting in the near future. Thereby, new genomic loci and types of pathogenic variants will be identified, resulting in more precise discrimination into different molecular subgroups. These subgroups serve as the basis for (epi)genotype–phenotype correlations, allowing a more directed prognosis, counseling, and therapy. By deciphering the pathophysiological consequences of SRS and BWS and their molecular disturbances, future therapies will be available targeting the basic cause of the disease and respective pathomechanisms and will complement conventional therapeutic strategies.

## 1. Introduction

Silver–Russell and Beckwith–Wiedemann syndromes (SRS, OMIM 180860; BWS, OMIM 130650) are rare congenital human disorders characterized by opposite growth disturbances and molecular alterations (Figure 1 and Figure 2, Table 1). Both syndromes have been assigned to the group of imprinting disorders, as the major molecular changes in both affect imprinted chromosomal regions.

SRS is associated with pre- and postnatal growth retardation, relative macrocephaly, prominent forehead, asymmetry, feeding difficulties, and further rather heterogeneous features [1]. The clinical severity of the condition varies widely between affected individuals. In adolescents and adults, the features may become less clear and clinical diagnosis more difficult. The published incidence of SRS ranges from 1:30,000 to 1:100,000 [1] but the disorder is probably more common than previously suggested.

BWS is also characterized by a variable phenotype that may include macroglossia, anterior abdominal wall defects (including exomphalos), pre- and/or postnatal overgrowth, neonatal hypoglycemia, hemihypertrophy, and predisposition to embryonal tumors (particularly Wilms tumor) [2]. Its incidence has been estimated as at least 1 in 10,500, but it may be higher [2,3].

Both syndromes are associated with alterations affecting the chromosomal region 11p15.5. This region harbors two clusters of imprinted genes, i.e., genes which are monoallelically expressed either from the maternal or paternal allele (Figure 2) [4]. These clusters are under the control of two discrete DNA elements called imprinting centers 1 and 2 (ICs). Imprinting centers regulate the imprinting marking of its subordinated chromosomal region. Among others, imprinting marks consist of methylation at specific cytosine residues, summarized as differentially methylated regions (DMRs). The mechanisms by which the ICs control imprinted genes are complex and involve the insulation of genes on one side of the IC from enhancers on the other, mediated by the insulator protein CTCF and higher-order chromatin interactions. Another mechanism comprises non-coding RNAs (ncRNA) that originate from the IC, targeting histone modifications in the surrounding genes. Both mechanisms involve a variety of epigenetic marks, including DNA methylation and histone modifications, but the hierarchy of and interactions between these modifications are not yet understood. The challenge is to understand these complex chains of interactions, beginning with differential methylation of an IC in the germline and ending with imprinting of many genes, often in a cell lineage dependent manner.

Like the majority of the other known imprinting disorders, SRS and BWS are associated with similar types of molecular alterations, comprising both genomic variants affecting the coding DNA sequence and structure of imprinted genes and the regulation regions, as well as altered methylation of the respective ICs [5]. Whereas, uniparental disomies (UPDs), copy number variations (CNVs) and single nucleotide variants (SNVs) in imprinted genes or regions represent classical genomic mutations, the so-called epimutations comprise DNA modifications without obvious alterations of the DNA sequence at the locus itself. However, all four variants alter the balanced expression of the imprinted genes. Recently, the discrimination of epimutations as primary and secondary has been suggested [6]. The latter are defined as altered imprinting marks indirectly caused by genomic alterations in chromosomal regions or factors which functionally interact with the DMR, whereas in case of primary epimutations the causes are not known.

The most common underlying molecular change in SRS in loss of methylation of the imprinting center 1 in chromosome 11p15 (IC1 LOM) in 40% to 50% of cases (Figure 2). In 10% of patients, maternal UPD of chromosome 7 (upd(7)mat) can be detected, in another 10% alterations of the imprinted region in chromosome 14q32 occur [1,7].

In more than 70% of BWS patients, molecular alterations of the imprinting centers 1 and 2 in 11p15 can be observed, with IC2 LOM and paternal UPD 11p15 (upd(11p15)pat) as the major findings (50% and 20%, respectively) [2].

For both disorders, it is assumed that the variants in 11p15.5 alter the expression and/or function of imprinted genes in 11p15.5. Due to their imprinting status and their role in intrauterine and postnatal growth, in particular, altered expression of *IGF2* and *CDKN1C* has been suggested to be associated with growth disturbances (Figure 3) [8,9], but final proofs are outstanding.

Despite the recent advances in deciphering the molecular basis of SRS, BWS, and other imprinting disorders, the mechanisms causing the known alterations as well as their pathophysiological consequences are widely unknown. Additionally, the current diagnostic approaches leave more than 40% of SRS and approximately 30% of BWS without molecular confirmation of their clinical diagnoses. These detection rates refer to patients exhibiting typical clinical features of SRS or BWS and are lower in patients with less characteristic features [10]. Furthermore, molecular testing for typical 11p alterations can result in unexpected and even opposite molecular findings (e.g., IC1 LOM in patients with BWS features or IC2 LOM in case of SRS features) (Table 1) [11].

This limited diagnostic yield in patients with features of SRS or BWS requires a broader molecular testing strategy, which now becomes possible with the implementation of high throughput techniques in the diagnostic workup. In fact, the precise molecular diagnosis in imprinting disorders is the prerequisite for a more precise therapy in terms of both conventional therapeutic strategies (e.g., for a specific growth treatment, tumor surveillance) and new personalized approaches.

In this review, we aim to review the current knowledge on pathobiology and pathophysiology of SRS and BWS as models for opposite congenital phenotypes associated with contrary molecular disturbances. By understanding the functional (epi)genotype–phenotype links and the affected physiological pathways, therapeutic strategies can be improved and developed.

## 2. Disturbed Imprinting is The Major Molecular Change in BWS and SRS Phenotypes

Up to now, molecular testing in SRS and BWS has been restricted to evidence-based molecular alterations, comprising disturbances of the imprinted regions in 11p15.5 and upd(7)mat in case of SRS. With the commercial availability of methylation-specific (MS) multiplex ligation-dependent probe amplification (MLPA) based tests targeting several imprinted loci on different chromosomes, the number of tested patients with the suspicious diagnosis of SRS or BWS has increased. However, there is a growing demand for molecular testing to confirm or exclude the clinical diagnosis. In fact, it should be emphasized that, as in other genetic disorders, exclusion of an imprinting disorder is hardly possible. This is due to the widespread mosaic distribution of epimutations in an individual, the targeted analysis of specific genomic regions leaving out others (e.g., introns), the unspecificity of symptoms and the genetic heterogeneity of SRS, BWS, and the other imprinting disorders.

In case of the 11p15.5 associated disorders, the application of MS MLPA allows the identification of three different molecular alterations (epimutations, UPD, CNVs), affecting the IC1 and IC2, in the same run [31]. Meanwhile, MS MLPA assays are available targeting all imprinted loci known to be associated with imprinting disorders [23].

As mentioned before, MS MLPA is suitable to identify the majority of characteristic molecular alterations in 11p15.5, but it also detects unexpected and even molecular opposite (epi)genetic variants. Hence, several molecular alterations which do not fit with clinical features of SRS or BWS but are consistent with the other 11p15.5 associated imprinting disorder, have been reported, e.g., patients referred as SRS with IC2 LOM or those diagnosed as BWS with IC1 LOM [11]. In these cases, the clinical reevaluation revealed that only one typical SRS or BWS symptom was present in these patients, and therefore, the diagnosis could not be confirmed. In summary, these cases show the clinical heterogeneity of SRS and BWS, and the need to apply appropriate genetic tests.

As recent data show, methylation testing in BWS and SRS should not be restricted to the 11p15.5 loci (and those on chromosome 7 in case of SRS), because a broad spectrum of alterations at imprinted loci others than in 11p15.5 can be detected in such patients. Two causes are responsible for this observation: (a) Overlap between clinical phenotypes of BWS, SRS, and other imprinting disorders, and (b) a significant number of patients with LOM of either IC1 or IC2 carry so-called multilocus imprinting disturbances (MLID), harboring varying genome-wide methylation alterations of imprinted loci.

(a) The overlapping phenotypes of imprinting disorders make a clear clinical classification sometimes difficult. Thus, it is not surprising that molecular diagnostics sometimes detect genomic alterations known for another imprinting disorder than the one which has been initially suspected (Table 1). In the case of BWS, a gain of methylation (GOM) of the MEG3 DMR in 14q32 has been reported, a finding typically associated with Kagami–Ogata syndrome [25]. Vice versa, LOM of the same DMR in 14q32 is detectable in a significant number of patients referred to as SRS [26]. Other overlapping molecular findings are 6q24 alterations, linking transient neonatal diabetes mellitus (TNDM) and BWS, and variants in 20q13 affecting the GNAS locus which are detectable in SRS [10,24].

(b) In the majority of imprinting disorder patients, only the disease-specific loci are affected, but an increasing number of individuals has been identified to carry MLID [32]. It is currently estimated that MLID mainly occurs in patients with chromosome 11p15.5 associated disorders (SRS, BWS), TNDM, and Pseudohypoparathyreoidism 1B (PHP1B). However, the ratio of MLID cases among epimutation carriers and its clinical relevance is unknown because MLID testing is based on analysis targeting either only a small number of loci or a limited number of CpGs within a DMR. Additionally, MLID mainly occurs as mosaic and might, therefore, escape detection [33]. Clinically, patients with MLID do not show significant clinical differences in comparison to patients with apparent isolated epimutations.

## 3. Genomic Alterations Resulting in SRS and BWS Phenotypes

In addition to the occurrence of disturbed imprinting patterns, there is a growing number of patients with molecular and clinical features of SRS or BWS exhibiting genomic alterations, i.e., insertions, deletions, duplications. Currently, three different groups of genomic variants can be discriminated:

### 3.1. Genomic Variants in 11p15.5 and Other Imprinted Regions Affecting the Function or the Expression of the Imprinted Genes in This Region

The development of high throughput and high-resolution DNA analyzing techniques (DNA microarray analysis, next and third generation sequencing (NGS/TGS)) has made the detailed characterization of the structure and sequence of the two ICs in 11p15 (and other regions) and the genes under their control possible [15]. By these approaches, the identification of central regulative elements became possible and provided insights into the regulation of imprinted genes expression [15,34,35,36]. Furthermore, with the identification of pathogenic variants in *IGF2* and *CDKN1C*, their central role in the pathophysiology of the 11p15.5 associated imprinting disorders has become obvious (Figure 3). Finally, recent data from the IC2 in 11p15.5 suggest that pathogenic variants in a gene (*KCNQ1* in IC2) do not only affect the function of the encoded protein itself but also the methylation of the respective DMR [13,34].

### 3.2. Variants in the Mothers of MLID Patients Causing Disturbed Imprinting Marks in Their Offspring

The second group of variants in families with imprinting disorders is associated with an uncommon mode of inheritance, the so-called maternal effect variants. This mode of inheritance describes the observation that the mothers in these families are carriers of pathogenic variants without clinical features but which cause aberrant imprinting in the offspring, i.e., MLID and LOM of the IC2 in 11p15.5 (Table 1). These variants affect genes encoding factors of the subcortical maternal complex, which mediate the proper setting of imprinting marks in the oocyte and early embryo (e.g., *NLRP2, NLRP5, NLRP7*; [37]). Maternal effect variants were first described in hydatidiform moles and other pregnancy complications [38], but it is, meanwhile, well established that they also cause aberrant methylation in imprinting disorders [39,40].

### 3.3. Variants Affecting Non-Imprinted Regions or Genes Resulting in Phenotypes Similar to SRS or BWS, i.e., Differential Diagnoses

With the application of whole exome sequencing (WES), whole genome sequencing (WGS), and DNA microarrays in genetic diagnostics of SRS and BWS, not only alterations within imprinted regions are detected, but also an increasing number of pathogenic genomic changes in other chromosomal regions could be identified. In fact, with *HMGA2* and *PLAG1,* new genes causing SRS have been found [18], thereby a new physiological pathway (HMGA2-PLAG1-IGF2-pathway) relevant for physiological of both fetal and postnatal growth could be identified. Furthermore, WES/WGS approaches will help to identify pathogenic variants in methylation-related transcription factors which might cause aberrant imprinting.

In future, these (epi)genome tests should be complemented by further *omic* approaches (e.g., transcriptomics) which will allow the parallel determination of the functional relevance of (epi)genetic alterations on the expression of some imprinted genes [41]. Thereby, networks of permissible physical interactions through which enhancers, promoters, insulators, and chromatin-binding factors can be identified which cooperatively regulate gene expression, reflected by the organization of accessible chromatin across the genome. First, data do not only unravel these functional cross-talks between imprinted loci (e.g., by lncRNAs [42]), this network of permissible physical interactions through which enhancers, promoters, insulators, and chromatin-binding factors cooperatively regulate gene expression is also reflected by the organization of accessible chromatin across the genome.

However, the majority of genomic variants are detected in genes associated with differential diagnosis of SRS and BWS [1,2,10,22]. The list of these variants and genes is rapidly growing, and it reflects the clinical overlap between both syndromes and other growth retardation or overgrowth disorders. However, it also illustrates the difficulty in the clinical diagnosis of these heterogeneous disorders and the unspecificity of their major features. Nevertheless, the precise molecular diagnosis is urgently required for a specific treatment (see below). Additionally, the clinical overlap between some of these disorders provides clues to the involvement of the different genes and chromosomal regions in common functional pathways (Figure 3).

One major reason to apply broad tests is the chance to identify the molecular causes in case of clinically and genetically heterogeneous disorders, as well as their cost efficiency and velocity in comparison to stepwise approaches. As a result, assays covering multiple genetic loci or even whole exomes (and genomes) are increasingly applied in routine diagnostics.

## 4. Imprinting Alterations Contribute to the Understanding of Its Underlying Mechanisms and Its Functional Relevance

The identification of pathogenic variants in imprinted chromosomal regions or genes causing aberrant imprinting as well as in genes associated with overlapping clinical pictures does not only increase the diagnostic yield but it also contributes to the understanding of the pathobiology of imprinting disorders and their functional consequences.

### 4.1. The Imprinted Genes Network

In the last years, it has become obvious that many imprinted genes interact in a so-called imprinted genes network (IGN) [41]. Evidences for the existence of the IGN are based on genomic, epigenomic, and transcriptomic datasets, associated gene function, and phenotype information. These data ultimately reflect coordinated mRNA synthesis via shared transcription factors (TFs), mRNA stability, mRNA translation, and protein stability.

As outlined earlier, there is considerable clinical and molecular overlap between different imprinting disorders, which can be explained by the IGN regulating zinc-finger TF PLAGL1. PLAGL1 is co-expressed with hundreds of genes, many of which are imprinted [43] among them *H19, IGF2*, and *CDKN1C* as members of the imprinting clusters in 11p15.5 [43,44]. Changes in the availability or activity of PLAGL1 or other TFs result in the altered expression of downstream targets, including these three genes and in line with this effects the SRS and BWS phenotypes.

This complex network of interactions between multiple genes and pathways is also highlighted by the role of noncoding RNAs encoded by the 11p15.5 region and other imprinted regions (e.g., miR-675, miR-483-3p) which act as microRNAs. In fact, the molecular function of these ncRNAs is often unclear, but a role for the attenuation of imprinted genes expression in specific cells or tissues [45] or in functional cross-talks between imprinted loci has been suggested [42]. Additionally, further networks have been described to regulate growth associated imprinted genes, e.g., the role of insulin and insulin-like growth factor 1 receptors [46]. This novel non-canonical mechanism is also independent of PLAGL1 despite influencing many of the same target genes, suggesting that the IGN can be subdivided into smaller interconnected units [47].

### 4.2. Multilocus Imprinting Disturbances

The coordinated control of imprinting at different loci is corroborated by the identification of multilocus imprinting disturbances (MLID) in a significant number of patients with specific imprinting disorders. This occurrence of aberrant methylation at different chromosomal loci indicates that common genomic alterations localized in trans are involved in the etiology of MLID. As preliminary data show, these alterations affect the maintenance of imprinting marks in the developing embryo, and the factors mediating this process are either expressed from the embryonal genome or are already expressed in the oocyte [37].

An example of a factor expressed from the embryonic genome is ZFP57, loss of function mutations occurs in TNDM cases with MLID [48]. In the mouse, Zfp57 associates preferentially with the ICs of Plagl1/Zac and Peg3, mirroring the loci commonly affected in human patients with *ZFP57* mutations [49,50,51].

As aforementioned, mutations in factors expressed in the oocyte also contribute to aberrant imprints and comprise maternal-effect variants. Several members of the subcortical maternal complex in the oocyte have been identified to carry maternal effect variants [39], with NLRP7 as the most prominent factor. NLRP7 is also mutated in women suffering from hydatidiform moles, a pregnancy outcome with paternalization of imprinting [52,53]. Maternal mutations of NLRP2 and NLRP5 have been associated with MLID in liveborn offspring [54,55]. Interestingly, the clinical features and epimutations in affected children are not consistent, suggesting that these mutations disrupt imprinting stochastically. Whereas, hydatidiform moles show extreme hypomethylation of maternal imprinting marks only, NLRP5 variants cause mosaic hypomethylation affecting both maternal and paternal marks, suggesting that NLRP7 may exert its effect in the oocyte, while NLRP5′s action may be postzygotic [37]. The inter-species variability and high similarity of genes in the NLRP cluster will pose challenges for dissecting their exact molecular roles in imprinting disorders.

Altogether, these observations contribute to the understanding of the mechanisms of imprinting regulation and its disturbances but also help to understand phenotypic outcomes, comorbidities, and clinical overlaps. Many imprinted genes are functionally related and part of common physiological pathways. These pathways and the IGN comprise both imprinted as well as non-imprinted networks, showing the complexity of the regulation of genomic imprinting.

### 4.3. Growth Disturbances As a Functional Result of Altered Imprinting

The major growth-related pathways affected in patients with 11p15.5 associated imprinting disorders include IGF2 (Figure 3). In fact, a growing number of molecular alterations affecting upstream factors of the IGF2 signaling pathway have been identified in SRS and BWS, including protein-coding and ncRNA encoding genes within imprinted and not imprinted regions. As depicted in Figure 3, they include the already known 11p15.5 imprinting disorder genes *IGF2, CDKN1C, HMGA2*, and *PLAG1*, but also genes mutated in patients with differential diagnoses of SRS or BWS (e.g., *IGF1R, DIS3L2*). In addition to the upstream factors, variants in genes downstream of IGF2 in the pathways are also associated with disturbed growth and features reminiscent of SRS and BWS [56]. An example is mutations in *PIK3R1*, which cause autosomal-dominant SHORT syndrome [57]. PIK3R1 encodes for a regulatory subunit of the Phosphatidylinositol 3-kinase (PI3K) and regulate, among other things, its catalytic activity. PI3Ks can be activated by IGF2-binding to the corresponding receptor and catalysis the phosphorylation of phosphatidylinositol (PI) [58]. These phosphorylated PIs serve as second messengers in the AKT-signaling pathway [59]. The activation of this pathway mediates protein biosynthesis, cell proliferation, and cell survival [60]. In addition to mutations in *PIK3R1*, there are also patients with activating mutations in the catalytic subunit PIK3CA of PI3K. These mutations lead to an overgrowth phenotype in contrast to patients with *PIK3R1* mutations resulting in growth retardation and reduced PI3K/AKT-signaling [61].

## 5. (Epi)Genotype–Phenotype Correlations in SRS and BWS

In parallel to the identification of the heterogeneous molecular alterations associated with SRS and BWS features, numerous studies aimed to determine the correlation between these changes and clinical features as the basis for a more directed therapy [62]. Though these studies are ongoing, at least for BWS, a strong correlation of molecular sub cohorts and the occurrence of clinical signs has been delineated [2]. For instance, alterations of the IC1 are associated with a higher risk for Wilms tumor and therefore, require another monitoring program other than changes of the IC2, which are rather linked to exomphalos. Additionally, recent publications indicate a functional link between *KCNQ1* variants predisposing to LongQT disorder and IC2 LOM [13]. Future studies will show whether this sub cohort should be cardiographically monitored.

In SRS, clinical differences are less obvious between the major molecular subgroups, IC1 LOM in 11p15, upd(7)mat and changes affecting the 14q32 DMRs [1,63,64], but have to be considered for clinical management. As growth retardation is one of the most relevant issues of patients with SRS features, growth hormone (GH) treatment is in the focus of treatment. In fact, several studies indicate that SRS patients generally benefit from GH treatment, but upd(7)mat patients have the greatest height gain [62,65]. Metabolic health under and after GH treatment seems to be the same in SRS as well as in children who are small for gestational age [66]. However, further data are needed to corroborate these findings and to allow more individualized therapy.

Particular care should be given to patients with 14q32 alterations, as [64] showed an earlier onset of puberty and an early increase in body mass index in comparison to other SRS subgroups. Additionally, early or premature adrenarche is reported to be more frequent in SRS than in the general population, associated with an early age at the initiation of GH treatment [62].

The cognitive development of patients with the clinical diagnosis of SRS and BWS is reported to be in the normal range in the majority of patients. However, specific molecular subtypes predispose developmental delay in some individuals. In SRS, neurocognitive problems are more frequent in patients with upd(7)mat [1] than in other subgroups. Thus, these patients need specific support. In carriers of CNVs affecting imprinted regions, the variants should be critically evaluated as the size and content of the affected chromosomal region(s) significantly influence the clinical outcome. As a result, the carriers might exhibit both developmental delay as well as dysmorphisms and malformations. In these cases, the clinical diagnosis of an imprinting disorder might even be complicated or undiagnosed due to the increased number of features.

## 6. The Molecular Diagnosis Is the Prerequisite for Precise Clinical Management and Counseling

Though SRS and BWS are, meanwhile, well-established entities and huge clinical datasets are available in the neonate and childhood period, systematic follow-up studies to estimate the impact on life-long health and quality of life are outstanding. In fact, the first studies on metabolism and psychosocial situation in adults with SRS have been published [67,68], but comprehensive surveys in adult patients are urgently required as the basis for directed multidisciplinary management. However, outcomes of the first studies advise the incorporation of psychological services in SRS and monitoring metabolism over time.

In addition to the need to precisely determine the molecular subtypes of chromosomes 7 and 11 disturbances, the identification of the molecular cause in patients with SRS or BWS features but without the characteristic (epi)mutation might also have a massive impact on the clinical management. Several differential diagnoses of SRS and BWS are associated with specific features requiring special attention and treatment [1,2]. Examples for differential diagnoses of SRS are Bloom syndrome or Mulibrey nanism where GH treatment is not indicated, due to the tumor predisposition associated with these syndromes.

In the future, understanding of the pathobiology of imprinting disorders will broaden the current conventional approaches by precise therapeutic strategies directly targeting the molecular cause of the disease. The first potential therapeutic interventions have been suggested for Angelman syndrome by reducing the UBE3A antisense transcript with antisense oligonucleotides (ASO) [69]. Experiences from other monogenetic disorders revealed that targeted oligonucleotide approaches are promising therapeutic strategies and have already been approved by the FDA [70]. Specific for imprinting disorders, epigenetic therapies based on methyltransferase inhibitors, histone deacetylase inhibitors, or “pro-methylation” dietary supplements might become efficient therapeutic options [71,72,73].

Finally, the identification of the molecular cause in patients with SRS or BWS is required to estimate the recurrence risk in their families. Whereas, the majority of epimutations in both disorders occurs sporadically and is, therefore, not associated with a recurrence probability, in particular, CNVs can be associated with a significantly increased recurrence risk of up to 50%. In the case of maternal-effect mutations, this risk can even be higher.

Furthermore, in context with the use of WES or WGS tests, it has to be considered that incidental findings might be identified. These variants are not functionally related to the disorder itself but might have a predictive value for other disorders. In fact, this aspect has to be discussed with the patients and their families before genetic testing. If possible, a molecular diagnostic workup should, therefore, rather start with targeted assays addressing disease-specific genetic loci to circumvent the detection of incidental findings and/or variants of unknown significance, which might cause uncertainty and anxiety in the patients and their families. Therefore, the use of diagnostic tools covering whole exomes or genomes should only be used after careful weighing of the pros and cons.

## 7. Conclusions and Outlook

With the increasing knowledge on the molecular basis of SRS and BWS, it has become obvious that the disorders mirror opposite alterations at the same genomic loci by contradictory phenotypic outcomes. In fact, these changes directly or indirectly affect the expression of IGF2 and its associated pathways and thereby, cause growth disturbances as key features of both diseases.

The increase of knowledge has become possible with the development and implementation of new and comprehensive assays. Whereas, in the beginning, molecular testing was restricted to single chromosomal loci, many tests now address numerous loci in the same run, and the diagnostic implementation of (epi)genome-wide assays is only a question of time. These high-throughput approaches will be complemented by the analysis of other *omic* datasets (e.g., transcriptome, metabolome, proteome), and it can be expected that the integration of these data will massively improve the understanding of the pathobiology of imprinting disorders and their diagnostics. Especially long-read sequencing methods, e.g., nanopore sequencing, allowing direct detection of native DNA modification, will strongly contribute to a better understanding of genomic imprinting in the near future. Thereby, new genomic loci and types of pathogenic variants could be identified, resulting in more precise discrimination into different molecular subgroups. These subgroups serve as the basis for (epi)genotype–phenotype correlations, allowing a more directed prognosis, counseling, and therapy. By deciphering the pathophysiological consequences of SRS and BWS and their molecular disturbances, future therapies will be available targeting the basic cause of the disease and respective pathomechanisms, and will thereby, complement conventional therapeutic strategies.

In addition to the benefit to the patients, these improvements will also support their families in decision making, by providing them with more information on genetic risks and therapeutic options.

## Figures and Tables

**Figure 1 ijms-20-04219-f001:**
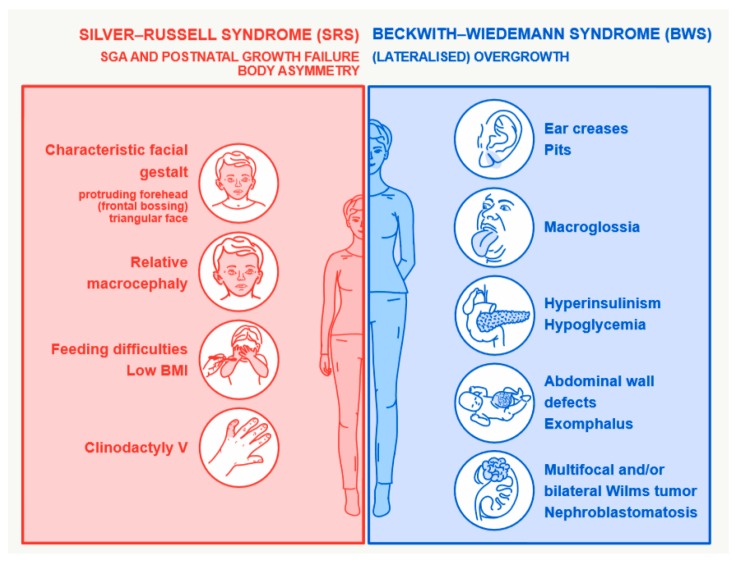
Major clinical features of Silver–Russell syndrome (SRS) and Beckwith–Wiedemann syndrome (BWS).

**Figure 2 ijms-20-04219-f002:**
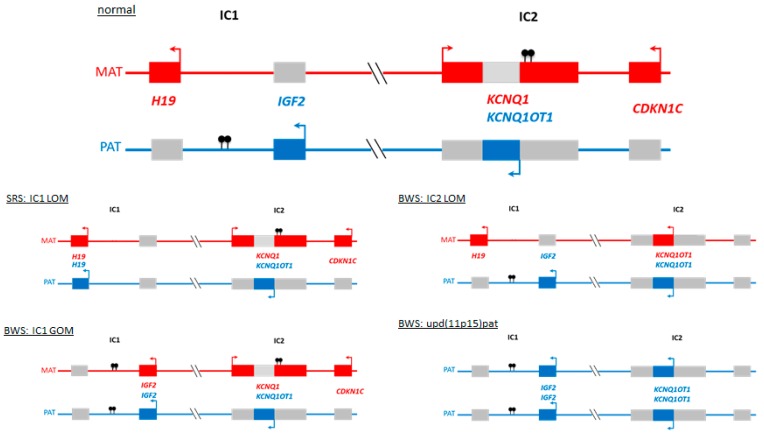
The imprinted 11p15.5 regions IC1 and IC2: normal situation and the major molecular alterations detectable in patients with SRS and BWS. (MAT maternal chromosome, red genes expressed from the maternal allele only; PAT paternal chromosome, blue genes expressed from the paternal allele only; grey silent gene copies; arrows indicate expressed allele; filled lollipops indicate methylation).

**Figure 3 ijms-20-04219-f003:**
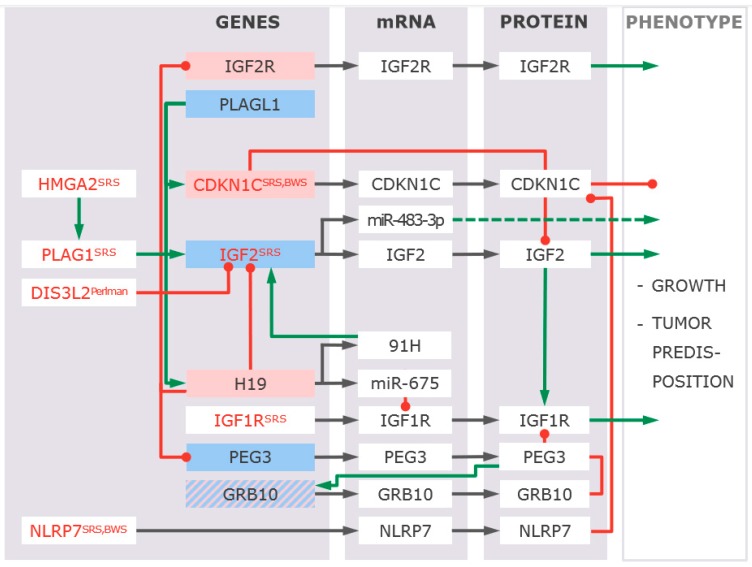
Interactions between genes pathogenic variants in which have been reported in SRS, BWS or related phenotypes (red letters, associated disorders are indicated). Functional interactions can be observed on expression as well as on physiological levels. Genes depicted in blue boxes are expressed from the paternal allele, whereas in red boxes maternally expressed genes are shown. Promoting interactions are indicated by green lines/arrows, red lines indicate inhibition. The effect of pathogenic variants in these genes/factors, such as over/underexpression, is not shown.

**Table 1 ijms-20-04219-t001:** Molecular findings in patients with Beckwith–Wiedemann syndrome (BWS) or Silver–Russell syndrome (SRS) features.

Genomic Region	Alteration	BWS/Overgrowth	SRS/Growth Retardation	Reference
	Alterations within imprinted regions			
11p15.5: IC1	GOM	5%–10% (in up to 20% SNPs/SNVs in transcription factors binding sites in the IC1 can be detected)	NR	[1,12]
	LOM	single cases *	40%–50%	[2,11]
11p15.5: IC2	LOM	50% (in some patients disturbance of the KCNQ1 transcript)	Single cases *	[1,11,13]
11p15.5: IC1 and IC2	Duplication	Maternal: <3%	Paternal: <1%	[14]
	UPD	upd(11)pat: 20%	upd(11)mat: 1 case	[1]
11p15.5: IC1 OR IC2	Small CNVs **	Single cases	Single cases	[15,16]
*11p15.5*	*CDKN1C*	LoF: 5% of sporadic, 40% of familial cases	GoF: single cases	[1,17]
	*IGF2*	NR	Familial and rare sporadic cases	[18,19]
Chromosome 7	UPD	upd(7)pat: 1 case	upd(7)mat: 7%–10%	[2,20]
	Segmental UPD7q		upd(7q)mat: single patients	[21]
	CNVs		Dup 7p13: GRB10 Del 7q32: MEST	[22]
Chromosome 6	UPD	upd(6)pat (TNDM)	upd(6)mat	[23,24]
Chromosome 14q32	Epimutation	MEG3 GOM (KOS14)	MEG3 LOM (TS14)	[25,26]
	CNVs **		Del14q32 (TS14) [26]	[26]
	UPD		upd(14)mat	[26]
Several imprinted regions	MLID ***	30% of IC2 LOM	15%–38% of IC1 LOM	[1,2]
Genomic variants in non-imprinted genes ***				
	*NSD1*	Del	Dup	[27,28]
	PIK3 function	*PIK3CA* overgrowth	*PIK3R1* SHORT	[29,30]

LOM—loss of methylation; GOM—gain of methylation; CNV—copy number variation; MLID—multilocus imprinting disturbance; LoF—loss of function; GoF—gain of function; * in these cases clinical features were not convincing for BWS or SRS, respectively; ** CNVs might affect the DMR itself or further non-coding areas which are directly or indirectly involved in the regulation of the imprinted region, e.g., KCNQ1OT1 in IC2; *** in fact, MLID might be caused by pathogenic variants in non-imprinted genes (see text).

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
