# Peer review of "Molecular and Clinical Opposite Findings in 11p15.5 Associated Imprinting Disorders: Characterization of Basic Mechanisms to Improve Clinical Management"

_ijms, 2019, doi:10.3390/ijms20174219_

Round 1

Reviewer 1 Report

The manuscript by T Eggermann, K Wesseler and F Kraft entitled “Molecular and clinical opposite findings in 11p15.5 2 associated imprinting disorders: Characterization of 3 basic mechanisms to improve clinical management” described the known molecular mechanisms of two pathologies.

In general the MS is well written and interesting. I have a few comments, however.

Figure legends should be self-explanatory. In particular Figure 2 has minimal descriptions e.g. blue/red, MAT/PAT, markers (I assume methylation), etc. This can be assumed by the text but the figure is before large part of the text describing it.

Figure 3 does not tell if the mutations result in activation or inactivation of the genes, or just over/underexpression.

The word “affecting” does not tell much as X might negatively or positively affect Y, or might indirectly/directly affect negatively/positively the structure/methylation/expression/repression of Y. If used as in a headline (Table 1) it requires explaining (arrows up/down could be useful for the sake of brevity, some explanations are in the Table legend such as LOM/GOM… but the issue remains, how is the loss or gain of methylation affecting the adjacent gene(s).

No info on changes in non-coding areas. There are plenty of good examples of changes in non-coding loci that affect the expression or near- and far-away genes. Even genes not in the cluster., potentially in a different chromosome.

Line 112-113: …therapeutic strategies (e.g. growth treatment…

What growth treatment means? Treatment using growth hormone? Or?

Line 194: c) With the application of WES, WGS and DNA microarrays

What are WES, WGS… no description of these abbreviations.

Line 196: … In fact, with HMGA2 and PLAG1 new genes causing SRS have been found [27],

Unclear line and no description to HMGA2/PLAG1

Lines 270-271: imprinted as well as non-imprinted networks, showing the complexity of genomic imprinting

I guess genomic structure: regulation is  a better term here

Line 287: PIK3R1, there are also patients with activating? Mutations

“?” why?

Virtually nothing is mentioned about the issue of testing gene expression in parallel to genome and epigenomic sequencing. This is of importance as merely methylations or variations might reduce/increase the expression of some genes while others might be able to compensate for those. Also SNP/SNV might reduce the impact of a TF but what if that TF is also “mutant” and perfectly functional to the SNP region? (purely theoretically but possible). Also how about gene expression in patients treated with e.g. GH? The differences between patients might and correlations to responsiveness might be of high importance to personalised medicine.

The use of methyltransferase inhibitor is shortly discussed but not the use of histone deacetylase inhibitors which could well be of interest as histone modifications are also reported in these 2 syndromes, as well as in other disorders including cancers.

Other methodologies such as those studying chromosomal cross-talk, cis and trans activation e.g. enhancers will bring new light to these diseases. This shall be described at least, if not yet done in these disorders, as a technological breakthrough in other diseases.

I recommend that the authors make a figure representing the opposite genetic alterations in these 2 syndromes. This will be values by the readers.

Minor details:

Line 12:.. SRS and BWS it… the abbreviations are not really described before despite that they are “obvious”.

Line 45: What means “ and less constant features”

Line 115: In the following, we…

The following what? Section? Paragraph?

Line 102: “…of SRS and BWS,…” should be “of SRS or BWS… as I’m assuming these patients do not suffer from both syndromes.

11p15.5

CDKN1C

LoF: 5% of sporadic, 40% of amilial cases

Should be “familial cases”

Line 136: Hence several molecular alterations does not fit with

Hence several molecular alterations do not fit with

Line 206: One major reason to apply broad tests are the chance

One is singular

Same line: to identify the molecular cause in

I would suggest to make it plural (causes).

Author Response

In general the MS is well written and interesting. I have a few comments, however.

Figure legends should be self-explanatory. In particular Figure 2 has minimal descriptions e.g. blue/red, MAT/PAT, markers (I assume methylation), etc. This can be assumed by the text but the figure is before large part of the text describing it.

Answer: We thank the reviewer for his suggestion and regret that we did not contribute a detailed legend in our first submission.

Figure 3 does not tell if the mutations result in activation or inactivation of the genes, or just over/underexpression.

Answer: We agree that an illustration of the functional effects of the variants would be helpful, but we feel that this would make the figure more complicated. Therefore we mentioned this in the legend now.

 The word “affecting” does not tell much as X might negatively or positively affect Y, or might indirectly/directly affect negatively/positively the structure/methylation/expression/repression of Y. If used as in a headline (Table 1) it requires explaining (arrows up/down could be useful for the sake of brevity, some explanations are in the Table legend such as LOM/GOM… but the issue remains, how is the loss or gain of methylation affecting the adjacent gene(s).

Answer: We agree that the term “affecting” in the table implicates to explain how the alterations affect the function of the related genes. This information  might be drawn from the cited literature, but we do not want to include it as this field is very complex and for some imprinted regions rather hypothetical than proven. We therefore changed the wording from “alterations affecting imprinted regions” to “alterations within imprinted regions”. Furthermore, we added information on MLID (“***”).

 No info on changes in non-coding areas. There are plenty of good examples of changes in non-coding loci that affect the expression or near- and far-away genes. Even genes not in the cluster., potentially in a different chromosome.

Answer: We have added a comment in table 1 (“**”) that changes in non-coding regions in  a cluster might affect the imprinting status, as well as alterations of genes in other chromosomes (MLID, “***”).

 Line 112-113: …therapeutic strategies (e.g. growth treatment…

What growth treatment means? Treatment using growth hormone? Or?

Answer: “for a specific” has been added.

Line 194: c) With the application of WES, WGS and DNA microarrays

What are WES, WGS… no description of these abbreviations.

Answer: abbreviations are introduced now.

Line 196: … In fact, with HMGA2 and PLAG1 new genes causing SRS have been found [27],

Unclear line and no description to HMGA2/PLAG1

Answer: The function of these genes/variants are explained now.

 Lines 270-271: imprinted as well as non-imprinted networks, showing the complexity of genomic imprinting

I guess genomic structure: regulation is  a better term here

Answer: Done

 Line 287: PIK3R1, there are also patients with activating? Mutations

“?” why?

Answer: The “?” was a mistake and has been removed.

Virtually nothing is mentioned about the issue of testing gene expression in parallel to genome and epigenomic sequencing. This is of importance as merely methylations or variations might reduce/increase the expression of some genes while others might be able to compensate for those.

Answer: This aspect has been added in paragraph 3.

Also SNP/SNV might reduce the impact of a TF but what if that TF is also “mutant” and perfectly functional to the SNP region? (purely theoretically but possible).

Answer: We have added a sentence in paragraph 3: “Furthermore, WES/WGS approaches will help to identify pathogenic variants in methylation-related transcription factors which might cause aberrant imprinting.”

Also how about gene expression in patients treated with e.g. GH? The differences between patients might and correlations to responsiveness might be of high importance to personalised medicine.

Answer: We have added a sentence for this topic in the second paragraph of the new chapter 5.

 The use of methyltransferase inhibitor is shortly discussed but not the use of histone deacetylase inhibitors which could well be of interest as histone modifications are also reported in these 2 syndromes, as well as in other disorders including cancers.

Answer: We thank the reviewer for this valuable comment and have added this idea in (new) section 6.

Other methodologies such as those studying chromosomal cross-talk, cis and trans activation e.g. enhancers will bring new light to these diseases. This shall be described at least, if not yet done in these disorders, as a technological breakthrough in other diseases.

Answer: We have added a sentence in paragraph 3.

I recommend that the authors make a figure representing the opposite genetic alterations in these 2 syndromes. This will be values by the readers.

Answer: This information is included in figure 2, but did not become clear due to the lack of an explanatory legend. This has been changed now (see first comment).

Minor details:

Line 12:.. SRS and BWS it… the abbreviations are not really described before despite that they are “obvious”.

Answer: Done.

Line 45: What means “ and less constant features”

Answer: The wording has been changed to: “further rather heterogeneous”

Line 115: In the following, we…

The following what? Section? Paragraph?

Answer: Changed to “in the following review”.

 Line 102: “…of SRS and BWS,…” should be “of SRS or BWS… as I’m assuming these patients do not suffer from both syndromes.

Answer: Changed.

11p15.5

CDKN1C

LoF: 5% of sporadic, 40% of amilial cases

Should be “familial cases”

 Answer: Done

Line 136: Hence several molecular alterations does not fit with

Hence several molecular alterations do not fit with

 Answer: Changed.

Line 206: One major reason to apply broad tests are the chance

One is singular

Answer: Changed.

Same line: to identify the molecular cause in

I would suggest to make it plural (causes).

 Answer: Changed.

In general the MS is well written and interesting. I have a few comments, however.

Figure legends should be self-explanatory. In particular Figure 2 has minimal descriptions e.g. blue/red, MAT/PAT, markers (I assume methylation), etc. This can be assumed by the text but the figure is before large part of the text describing it.

Answer: We thank the reviewer for his suggestion and regret that we did not contribute a detailed legend in our first submission.

Figure 3 does not tell if the mutations result in activation or inactivation of the genes, or just over/underexpression.

Answer: We agree that an illustration of the functional effects of the variants would be helpful, but we feel that this would make the figure more complicated. Therefore we mentioned this in the legend now.

 The word “affecting” does not tell much as X might negatively or positively affect Y, or might indirectly/directly affect negatively/positively the structure/methylation/expression/repression of Y. If used as in a headline (Table 1) it requires explaining (arrows up/down could be useful for the sake of brevity, some explanations are in the Table legend such as LOM/GOM… but the issue remains, how is the loss or gain of methylation affecting the adjacent gene(s).

Answer: We agree that the term “affecting” in the table implicates to explain how the alterations affect the function of the related genes. This information  might be drawn from the cited literature, but we do not want to include it as this field is very complex and for some imprinted regions rather hypothetical than proven. We therefore changed the wording from “alterations affecting imprinted regions” to “alterations within imprinted regions”. Furthermore, we added information on MLID (“***”).

 No info on changes in non-coding areas. There are plenty of good examples of changes in non-coding loci that affect the expression or near- and far-away genes. Even genes not in the cluster., potentially in a different chromosome.

Answer: We have added a comment in table 1 (“**”) that changes in non-coding regions in  a cluster might affect the imprinting status, as well as alterations of genes in other chromosomes (MLID, “***”).

 Line 112-113: …therapeutic strategies (e.g. growth treatment…

What growth treatment means? Treatment using growth hormone? Or?

Answer: “for a specific” has been added.

Line 194: c) With the application of WES, WGS and DNA microarrays

What are WES, WGS… no description of these abbreviations.

Answer: abbreviations are introduced now.

Line 196: … In fact, with HMGA2 and PLAG1 new genes causing SRS have been found [27],

Unclear line and no description to HMGA2/PLAG1

Answer: The function of these genes/variants are explained now.

 Lines 270-271: imprinted as well as non-imprinted networks, showing the complexity of genomic imprinting

I guess genomic structure: regulation is  a better term here

Answer: Done

 Line 287: PIK3R1, there are also patients with activating? Mutations

“?” why?

Answer: The “?” was a mistake and has been removed.

Virtually nothing is mentioned about the issue of testing gene expression in parallel to genome and epigenomic sequencing. This is of importance as merely methylations or variations might reduce/increase the expression of some genes while others might be able to compensate for those.

Answer: This aspect has been added in paragraph 3.

Also SNP/SNV might reduce the impact of a TF but what if that TF is also “mutant” and perfectly functional to the SNP region? (purely theoretically but possible).

Answer: We have added a sentence in paragraph 3: “Furthermore, WES/WGS approaches will help to identify pathogenic variants in methylation-related transcription factors which might cause aberrant imprinting.”

Also how about gene expression in patients treated with e.g. GH? The differences between patients might and correlations to responsiveness might be of high importance to personalised medicine.

Answer: We have added a sentence for this topic in the second paragraph of the new chapter 5.

 The use of methyltransferase inhibitor is shortly discussed but not the use of histone deacetylase inhibitors which could well be of interest as histone modifications are also reported in these 2 syndromes, as well as in other disorders including cancers.

Answer: We thank the reviewer for this valuable comment and have added this idea in (new) section 6.

Other methodologies such as those studying chromosomal cross-talk, cis and trans activation e.g. enhancers will bring new light to these diseases. This shall be described at least, if not yet done in these disorders, as a technological breakthrough in other diseases.

Answer: We have added a sentence in paragraph 3.

I recommend that the authors make a figure representing the opposite genetic alterations in these 2 syndromes. This will be values by the readers.

Answer: This information is included in figure 2, but did not become clear due to the lack of an explanatory legend. This has been changed now (see first comment).

Minor details:

Line 12:.. SRS and BWS it… the abbreviations are not really described before despite that they are “obvious”.

Answer: Done.

Line 45: What means “ and less constant features”

Answer: The wording has been changed to: “further rather heterogeneous”

Line 115: In the following, we…

The following what? Section? Paragraph?

Answer: Changed to “in the following review”.

 Line 102: “…of SRS and BWS,…” should be “of SRS or BWS… as I’m assuming these patients do not suffer from both syndromes.

Answer: Changed.

11p15.5

CDKN1C

LoF: 5% of sporadic, 40% of amilial cases

Should be “familial cases”

 Answer: Done

Line 136: Hence several molecular alterations does not fit with

Hence several molecular alterations do not fit with

 Answer: Changed.

Line 206: One major reason to apply broad tests are the chance

One is singular

Answer: Changed.

Same line: to identify the molecular cause in

I would suggest to make it plural (causes).

 Answer: Changed.

Reviewer 2 Report

This interesting article revise the molecular features of imprinting diseases, highlighting their diagnosis difficulties. It is pointed out, how next omics techniques may help to overcome the apparent inconsistency between the clinical and the molecular data,  due to the genetic and epigenetic complexity of imprinting diseases. The manuscript is accurate and generally well written and deserves publication.

Author Response

Reviewer 2

This interesting article revise the molecular features of imprinting diseases, highlighting their diagnosis difficulties. It is pointed out, how next omics techniques may help to overcome the apparent inconsistency between the clinical and the molecular data,  due to the genetic and epigenetic complexity of imprinting diseases. The manuscript is accurate and generally well written and deserves publication.

Answer: We thank the reviewer for reviewing and the positive comments.

Reviewer 3 Report

In this manuscript, Eggermann et al. reviewed molecular mechanisms of two imprinting disorders, Silver-Russell and Beckwith-Wiedemann syndrome, and discussed outlooks for molecular diagnosis and clinical managements. This reviewer thinks that this manuscript is interesting and useful to the readers of International Journal of Molecular Sciences, however, there are several concerns.

1) The writing structure of the manuscript is not good. This reviewer thinks that it is hard to follow for the readers. For example, lines 142-164 in page 6, short summaries indicated as (a) and (b) were followed by detailed descriptions of (a) and (b). Section 3 and 4 were too long. The authors should reconstruct the manuscript. Especially, the authors should use “subsections” and “subsubsections” according to “ijms-template”.

2) Lines 58–68, page 3: DMRs are not always CpG islands. Especially, IC1 is not a CpG island. Did the authors want to describe IC1 and IC2 specifically, or general issues of ICs?

3) Line 72, page 3: It is better to use ICs instead of DMRs.

4) Lines 76-77, page 3: “Recently, the discrimination of epimutations as primary and secondary has been suggested.” Please add adequate references.

5) Lines 181 and 202, page 7: (see below) Where should the readers see?

6) Lines183-185, page 7 and lines 297-298, page 9: These two sentences seem to be inconsistent each other.

7) Lines 189-193, page 7: Description of this part is inadequate. Thus, it is hard to understand.

8) Lines 198, page 7: It is better to show the list of the representative variants.

9) Lines 210-217, page 7: A topic of this part seems to be a genetic counseling. Therefore, this part should be moved to Section 4.

10) Lines 274-276, page 8: Please add adequate references.

11) Line 287, page 9: What does “?” in “activating? mutation” mean?

12) Line 307, page 9: Author name should be used instead of the reference number.

13) Detailed discerptions are necessary for the legends of Figure 2 and 3.

14) Table 1 is hard to see and understand. The authors should make it easy to understand. 11p15.5 should be “variants of imprinted genes”. Why segmental UPD7q is a blank?

Author Response

The writing structure of the manuscript is not good. This reviewer thinks that it is hard to follow for the readers. For example, lines 142-164 in page 6, short summaries indicated as (a) and (b) were followed by detailed descriptions of (a) and (b). Section 3 and 4 were too long. The authors should reconstruct the manuscript. Especially, the authors should use “subsections” and “subsubsections” according to “ijms-template”.

Answer: The structure has been modified by introducing further headings.

2) Lines 58–68, page 3: DMRs are not always CpG islands. Especially, IC1 is not a CpG island. Did the authors want to describe IC1 and IC2 specifically, or general issues of ICs?

Answer: We have changed this sentence.

3) Line 72, page 3: It is better to use ICs instead of DMRs.

Answer: Changed.

4) Lines 76-77, page 3: “Recently, the discrimination of epimutations as primary and secondary has been suggested.” Please add adequate references.

Answer: Done.

5) Lines 181 and 202, page 7: (see below) Where should the readers see?

Answer: We refer now to reference:  F. Kraft, K. Wesseler, M. Begemann, I. Kurth, M. Elbracht, and T. Eggermann, “Novel familial distal imprinting centre 1 (11p15.5) deletion provides further insights in imprinting regulation.,” Clin. Epigenetics, vol. 11, no. 1, p. 30, Feb. 2019.

6) Lines183-185, page 7 and lines 297-298, page 9: These two sentences seem to be inconsistent each other.

Answer: These parts have been changed/restructured according to reviewer.

7) Lines 189-193, page 7: Description of this part is inadequate. Thus, it is hard to understand.

Answer: We have modified this part.

8) Lines 198, page 7: It is better to show the list of the representative variants.

Answer: We have added NLRP2, NLRP5 and NLRP7 as examples.

9) Lines 210-217, page 7: A topic of this part seems to be a genetic counseling. Therefore, this part should be moved to Section 4.

Answer: The section has been moved to the new paragraph 6.

10) Lines 274-276, page 8: Please add adequate references.

Answer: A reference has been added.

11) Line 287, page 9: What does “?” in “activating? mutation” mean?

Answer: Removed, we regret that mistake.

12) Line 307, page 9: Author name should be used instead of the reference number.

Answer: We considered this, but we find it difficult to introduce a second type of citation.

 13) Detailed discerptions are necessary for the legends of Figure 2 and 3.

Answer: The legends have been modified.

14) Table 1 is hard to see and understand. The authors should make it easy to understand. 11p15.5 should be “variants of imprinted genes”. Why segmental UPD7q is a blank?

Answer: We have modified the table, the gap in the segmental UPD7 line has been filled.

Round 2

Reviewer 3 Report

The authors revised the manuscript according to comments of the referees. However, the authors’ responses are insufficient and there are still some concerns.

1) The writing structure of “section 3” was not changed adequately. Subsections should be used instead of (a), (b), and (c).

2) Lines 63, page 3: This reviewer regrets to have to say again, “DMRs or ICs are not always CpG islands.”

6) Lines 194-196, page 7 and lines 343-345, page 10: This reviewer thinks that these two sentences have not been changed.

8) The authors added NLRP2, NLRP5 and NLRP7 as examples. However, the sentence, which this reviewer had pointed out, was another. The authors might get the wrong idea. Pease see the line number in the previous comment #8.

10) Lines 319-321, page 8: A reference has not been added.

Author Response

1) The writing structure of “section 3” was not changed adequately. Subsections should be used instead of (a), (b), and (c).

Answer: We are sorry for the misunderstanding and have changed from a/b/c to 3.1 etc. (marked in yellow)

2) Lines 63, page 3: This reviewer regrets to have to say again, “DMRs or ICs are not always CpG islands.”

Answer: we have hopefully changed this part in better way (marked in yellow).

6) Lines 194-196, page 7 and lines 343-345, page 10: This reviewer thinks that these two sentences have not been changed.

Lines 194-196:

Answer: We hope we refer to the correct lines: We have added MLID and IC2 LOM and hope that we have addressed the comment now.

Lines 343-345:

Answer: We hope we refer to the correct lines, and have modified the paragraph according to the reviewers ideas.

8) The authors added NLRP2, NLRP5 and NLRP7 as examples. However, the sentence, which this reviewer had pointed out, was another. The authors might get the wrong idea. Pease see the line number in the previous comment #8.

Answer: see 6) Lines 194-196.

10) Lines 319-321, page 8: A reference has not been added.

Answer: Current reference 52 is appropriate and has been added.